# BAT26 Only Microsatellite Instability with High Tumor Mutation Burden—A Rare Entity Associated with PTEN Protein Loss and High PD-L1 Expression

**DOI:** 10.3390/ijms231810730

**Published:** 2022-09-14

**Authors:** So Young Kang, Deok Geun Kim, Kyoung-Mee Kim

**Affiliations:** 1Department of Pathology and Translational Genomics, Samsung Medical Center, Sungkyunkwan University School of Medicine, Seoul 06351, Korea; 2Department of Clinical Genomic Center, Samsung Medical Center, Seoul 06351, Korea; 3Department of Digital Health, Samsung Advanced Institute of Health Science and Technology, Sungkyunkwan University, Seoul 06351, Korea; 4Center of Companion Diagnostics, Samsung Medical Center, Seoul 06351, Korea

**Keywords:** microsatellite instability, BAT26, tumor mutation burden, PTEN, PD-L1, next-generation sequencing

## Abstract

Detecting microsatellite instability (MSI) in advanced cancers is crucial for clinical decision-making, as it helps in identifying patients with differential treatment responses and prognoses. *BAT26* is a highly sensitive MSI marker that defines the mismatch repair (MMR) status with high sensitivity and specificity. However, isolated *BAT26*-only instability is rare and has not been previously reported. Of the 6476 cases tested using pentaplex MSI polymerase chain reaction, we identified two *BAT26*-only instability cases (0.03%) in this study. The case #1 patient was diagnosed with endometrial adenocarcinoma without MMR germline mutations. The endometrial tumor showed *BAT26*-only instability, partial loss of MLH1/PMS2 protein expression, and a high programmed cell death ligand 1 (PD-L1) combined positive score (CPS = 8). The tumor exhibited a somatic phosphatase and tensin homolog (*PTEN*) R303P missense mutation and loss of the PTEN protein. On a comprehensive cancer panel sequencing with ≥500 genes, the tumor showed an MSI score of 11.38% and high tumor mutation burden (TMB) (19.5 mt/mb). The case #2 patient was diagnosed with colorectal carcinoma with proficient MMR and PTEN protein loss without *PTEN* alteration, as well as a high PD-L1 CPS (CPS = 10). A pathogenic *KRAS* A146T mutation was detected with an MSI score of 3.36% and high TMB (13 mt/mb). In conclusion, *BAT26*-only instability is very rare and associated with PTEN protein loss, high TMB, and a high PD-L1 score. Our results suggest that patients with *BAT26*-only instability may show good responses to immunotherapy.

## 1. Introduction

Microsatellite instability-high (MSI-H) is a well-recognized phenomenon resulting from defective mismatch repair (dMMR) [1]. Low-level MSI (MSI-L) varies among different cancer types and has not been clearly defined in any group [2]. For MSI testing, marker selection, marker performance, and the percentage of markers showing instability in distinguishing microsatellite stable (MSS)/MSI-L from MSI-H are important issues when determining the MSI test results [3].

The mononucleotide marker *BAT26* (a poly(A) tract localized in the fifth intron of the mutS homolog (*MSH*)-*2*) [4] is an extremely sensitive and specific quasi-monomorphic marker for testing MSI. The use of mononucleotide markers alone has led to the correct identification of 97% of MSI-H cases, and *BAT26* is specific and sensitive in identifying MSI-H cases [5,6]. Cicek et al. reported that 91% of MSI-H cases demonstrated instability at the *BAT26* locus, and the *BAT26*-only MSI test showed 94% sensitivity and 98% specificity for the identification of tumors with MMR deficiency, suggesting *BAT26* as the best single marker to identify tumors with MSI-H [7]. In our institute, the MSI polymerase chain reaction (PCR) test was performed for 6476 cases, and *BAT26*-only instability was identified in two cases (0.03%), which is very rare, and there have been no previous reports of such cases in the literature.

Herein, we report two rare *BAT26*-only instability cases associated with high tumor mutation burden (TMB), phosphatase and tensin homolog (PTEN) protein loss, and high PD-L1 expression via a comprehensive cancer panel test and immunohistochemistry.

## 2. Results

### 2.1. Case Report #01

A 61-year-old woman visited the hospital for an operation of asymptomatic grade 2 endometrial endometrioid carcinoma incidentally diagnosed during regular health checks. A total hysterectomy with bilateral salpingo-oophorectomy and pelvic lymph node dissection were performed. The tumor invaded less than one half of the myometrium with no metastasis in any of the 10 regional lymph nodes (pT1aN0M0). The endometrial tumor showed *BAT26*-only instability with a partial loss of MLH1/PMS2 expression and a high PD-L1 combined positive score (CPS = 8). The tumor showed an “immune-excluded” microenvironment (immune cells aggregating at the tumor boundaries). NGS revealed the *PTEN* R303P alteration, while IHC demonstrated the PTEN protein loss (Figure 1). Cooccurring genetic alterations included *KRAS* (G12A) and AT-rich interaction domain 1A (p.Q1519Pfs*13 and p.D1850Tfs*33) mutations (Appendix A). The tumor showed an MSI score of 11.38%, along with a high TMB (19.5 mt/mb). In NGS, the microsatellite (MS) loci, including EWS RNA-binding protein 1, paired box 8, and *MYB*, also showed instability in endometrial tumors (Appendix A). The patient had two first-degree relatives with endometrial and breast cancers, respectively. The patient was previously diagnosed with colorectal cancer eight years ago, and the colon tumor was MSS and proficient MMR (pMMR). The clinician provided genetic counseling based on the patient’s cancer diagnosis and family history. After obtaining consent, a genetic analysis of the patient’s peripheral blood was performed via whole-genome sequencing, and no MMR germline mutation was found; thus, the patient was diagnosed with Lynch-like syndrome.

### 2.2. Case Report #02

A 48-year-old woman visited the emergency department due to hematochezia. An urgent colonoscopy detected a mass in the sigmoid colon, and a biopsy confirmed an adenocarcinoma. Of the 5903 colorectal cancer cases tested with MSI PCR, this case was the only one (0.017%) with *BAT26*-only instability. After the cancer diagnosis, a low anterior resection was performed. A histopathological examination of the surgically resected sample revealed moderately differentiated adenocarcinoma. Pathological staging identified the cancer as stage IIIA with invasion into the muscularis propria (pT2) and regional lymph node metastasis to 3–21 regional lymph nodes (pN1b). The tumor was pMMR with a high PD-L1 CPS score (CPS = 10) and an immune-excluded tumor microenvironment. The tumor showed a loss of the PTEN protein without a *PTEN* alteration (Figure 2).

The cooccurring genetic alterations were pathogenic *KRAS* A146T mutations. The MSI score was 3.36%, and the tumor showed high TMB (13 mt/mb). The *EPCAM* R153T variant from ClinVar was annotated as nonpathogenic but showed an intact EPCAM protein expression. Two variants (Q419K and Q629R) were classified as benign or likely benign within *MSH2*. Two variants (P18L and G25D) of uncertain clinical significance (VUS) were observed in mutY DNA glycosylase. The mononucleotide MS markers at *MYB* and lysosomal associated membrane protein 1 loci were unstable (Appendix A).

## 3. Discussion

The MSI status was determined using a standard panel of MS markers, as defined by the National Cancer Institute/Bethesda consensus guidelines [8]. Tumors with frameshifts in 30% or more of the marker genes were classified as MSI-high tumors [9]. Owing to its high sensitivity (94%) and specificity (98%), *BAT26* was initially proposed as an unambiguous single marker of MSI [6] and therefore included in both MS panels proposed at the first [1] and second [10] Bethesda consensus meetings. Brennetot et al. reported that *BAT26*, together with *BAT25*, accurately detects MSI-high tumors and can be used to predict the percentage of tumor cells in DNA samples [11]. Pastrello et al. reported that MSI-H with *BAT26*-only stability showed a large intragenic *MSH2* deletion, but MSI-L with *BAT26*-only instability was not observed in tumor DNA [12].

To the best of our knowledge, *BAT26*-only instability has not been previously reported in solid tumors. Notably, we identified two *BAT26*-only instability cases with high TMB associated with PTEN protein loss and high PD-L1 scores.

Using targeted comprehensive genomic profiling of 62,150 cancer samples, Chalmers et al. found that MSI-H is closely associated with gastrointestinal and gynecologic cancers [13]. Most MSI-H samples exhibited high TMB (83%), although the opposite was not true. In gastrointestinal and gynecological cancers, MSI-H and high TMB almost always cooccur, whereas, in melanoma, squamous cell carcinoma, and lung carcinoma, high TMB is fairly common, but MSI-H is very uncommon. These data suggest that the MSI-H pathway appears to be a common phenomenon in gastrointestinal and gynecologic carcinogenesis but is less frequently involved in other cancer types [14]. Chang et al. reported that important clinical implications of MSI testing in gastrointestinal and gynecological cancers may provide useful information for immunotherapy [14].

In the present study, we found *BAT26*-only instability in colorectal and endometrial carcinomas where MSI-H is commonly observed. One patient with PTEN inactivation showed *PTEN* mutation. Although this was a retrospective study and the number of patients with *BAT26*-only instability was low owing to its rarity, we demonstrated PTEN protein loss, high TMB, and high PD-L1 expression in these rare cases. We previously reported that *PTEN* mutations are frequently observed in MSI-H/dMMR tumors [15]. The scope of PTEN functions has recently expanded with new potential implications for immunotherapy-based approaches [16]. PTEN protein loss and high TMB are frequently observed in immune-excluded tumor microenvironments [17], and our findings are consistent with those of prior studies [18]. The present study also supports the view that *BAT26*-only instability arises largely due to the basal instability level of cancer genomes rather than representing a distinct disease category, such as MSI-L tumors [19]. Out of 6476 cases tested for pentaplex MSI-PCR, we identified two *BAT26*-only instability cancers. Unexpectedly, those two cases were associated with high TMB and high PD-L1 expression. Given that high TMB and PD-L1 overexpression are predictive biomarkers for immunotherapy [20], we suggest that, although very rare, patients with *BAT26*-only instability cancer need further tests for TMB and PD-L1 and would be a good candidate for immunotherapy.

As an intermediate between MSI-H and MSS cases, MSI-L tumors are thought to comprise 2–10% of colorectal cancers [21]. However, whether these genomes represent a unique disease entity that can be clearly separated from the MSI-H and MSS genomes with distinct clinical or genetic features has to be further elucidated [22]. Kim et al. reported that MSI-L arises due to the basal instability level of cancer genomes rather than representing a distinct disease category [19,23], and MSI-L cases may represent the pre-MSI-H phase during cancer progression. Although previous studies have demonstrated MSI-L tumors with the use of an extended set of MS markers (Appendix A), *BAT26*-only instability, which should be classified as MSI-L by its definition, showed somewhat different characteristics compared to MSI-L tumors. Giannini et al. reported that MSI events near splicing sites may alter the transcript level or splicing pattern of target genes, as shown for *MRE11* [24]. Zhou et al. previously reported that the quasi-monomorphic allelic nature of *BAT26* in the normal population has the potential functional significance of MS repeats around splice sites [25]. Our two *BAT26*-only MSI-L mutants showed poly-T/A repeats of the *MYB* MS marker, which is frequently unstable in *MLH1* and *MSH2* mutation carriers [26,27]. The current cases showed no MMR germline mutations.

In summary, we described two rare cases of *BAT26*-only instability associated with high TMB, PTEN protein loss, and high PD-L1 CPS, suggesting that patients with *BAT26*-only instability are good candidates for immunotherapy.

## 4. Materials and Methods

### 4.1. Patient Acquisition

Of the 6476 cases between December, 2017 and February, 2020, we identified two cases with *BAT26*-only instability based on the pentaplex MSI PCR results in the Department of Pathology of the Samsung Medical Center. This study was performed in accordance with the Institutional Review Board guidelines (IRB 2020-06-045-001) for the data analysis and review of the medical records.

### 4.2. MSI Test

The tumor-rich areas were circled by pathologists on hematoxylin and eosin-stained 4-μm-thick slides of formalin-fixed, paraffin-embedded tissues for dissection. The MSI status was determined via multiplex PCR, which amplified five quasi-monomorphic mononucleotide repeat markers (pentaplex: BAT25, BAT26, NR21, NR24, and NR27), as previously described [28,29].

### 4.3. Immunohistochemistry (IHC)

IHC staining was performed on paraffin-embedded 3-µm-thick tissue sections for MSH2, mutL homolog 1 (MLH1), MSH6, PMS1 homolog 2 (PMS2), epithelial cell adhesion molecule (EPCAM), programmed cell death ligand 1 (PD-L1), and PTEN [15,30]. An IHC analysis was performed using a Ventana Bench Mark XT autoimmunostainer (Ventana Medical Systems, Tucson, AZ, USA) after incubation with the primary antibodies at 37 °C for 30 min, followed by standard Ventana signal amplification and counterstaining with hematoxylin for 4 min, as previously described [15,30].

### 4.4. Next-Generation Sequencing (NGS) with Comprehensive Cancer Panel

Library preparation was performed using the hybrid capture-based Illumina TruSight Oncology 500 (TSO500; Illumina, San Diego, CA, USA), as previously described [30]. Data outputs exported from the TSO 500 pipeline (Illumina) were annotated using the Ensembl Variant Effect Predictor Annotation Engine [31] with information from several databases and alignment to the hg19 human reference genome (http://genome.ucsc.edu/) (accessed on 1 March 2022). Mutation allele frequencies below the predefined thresholds were considered to be wildtypes.

## Figures and Tables

**Figure 1 ijms-23-10730-f001:**
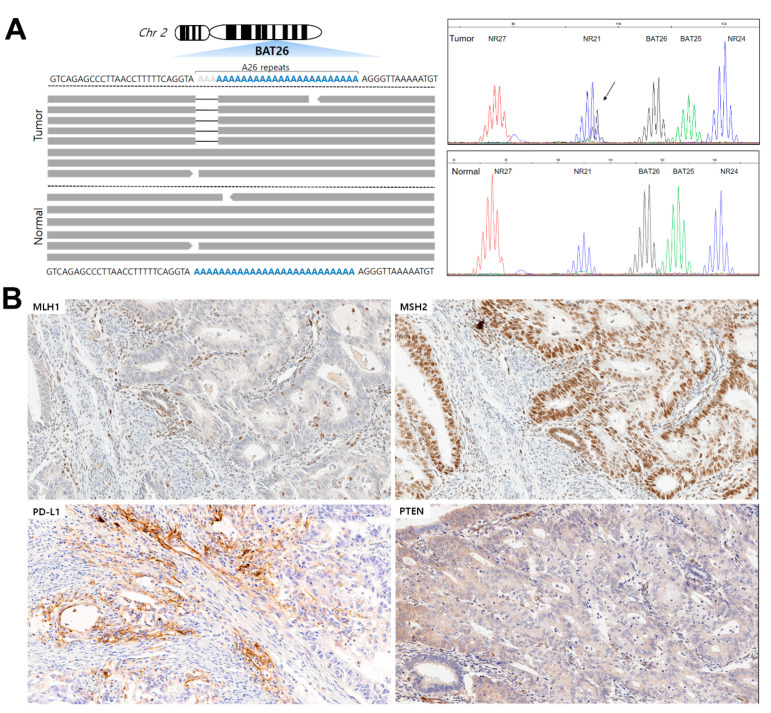
(**A**) In *BAT26*-only instability cancer, a mononucleotide (**A**) repeat was deleted by erroneous replication from the 3′ region of *MSH2* exon 5 on chromosome 2. Amplification profile from case #01 using pentaplex PCR showing a *BAT26*-only unstable phenotype in tumor tissue. (**B**) Immunohistochemistry of tumor section from case #01 using antibodies targeting MLH1 and MSH2, showing the expression of mismatch repair proteins in tumor cells (partial mismatch repair deficiency), PD-L1 expression in lymphocytes (immune-excluded tumor microenvironment; CPS = 8), and PTEN protein loss in tumor sections. All images are at 2.0× magnification.

**Figure 2 ijms-23-10730-f002:**
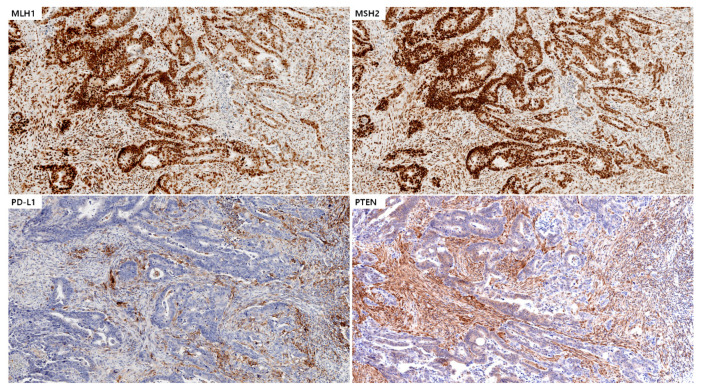
Immunohistochemistry of tumor sections from case #02 using antibodies targeting MLH1 and MSH2 showing the expression of mismatch repair proteins in tumor cells (mismatch repair proficiency), PD-L1 expression in lymphocytes (immune-excluded tumor microenvironment; CPS = 10), and PTEN protein loss in tumor sections. All images are at 2.0× magnification.

## Data Availability

The data are available upon reasonable request. The data that support the findings of this study are available upon request from the corresponding author, K.-M.K.

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
