# Peer review of "BAT26 Only Microsatellite Instability with High Tumor Mutation Burden—A Rare Entity Associated with PTEN Protein Loss and High PD-L1 Expression"

_ijms, 2022, doi:10.3390/ijms231810730_

Round 1

Reviewer 1 Report

This paper presents 2 patients who have BAT26-only MSI with high TMB, PTEN protein loss, and high PD-L1 expression. This report will be useful to many readers because it shows the frequency of instability of BAT26 alone and the associated genetic and immunological features. This topic is interesting and deserves a constructive discussion.

However, I concern some issues as listed below for publication.

I hope that my comment is useful for the improvement of the article.

Major concerns:

1.       In these case reports, it is noted that the both tumors were adenocarcinomas, but it would be better to write clearly a more detailed classification for understanding the cases. For example, in the case #01, it could be endometrioid carcinoma, serous carcinoma, clear cell carcinoma, etc.

Minor concerns:

1.       The authors should describe the occasions and first symptoms of the cancers in these patients when the cancers were discovered.

2.       Has the Case #01 patient been searched for any genetic conditions, including Cowden syndrome/PTEN hamartoma tumor syndrome?

Author Response

This paper presents 2 patients who have BAT26-only MSI with high TMB, PTEN protein loss, and high PD-L1 expression. This report will be useful to many readers because it shows the frequency of instability of BAT26 alone and the associated genetic and immunological features. This topic is interesting and deserves a constructive discussion.

However, I concern some issues as listed below for publication.

I hope that my comment is useful for the improvement of the article.

Major concerns:

  1. In these case reports, it is noted that the both tumors were adenocarcinomas, but it would be better to write clearly a more detailed classification for understanding the cases. For example, in the case #01, it could be endometrioid carcinoma, serous carcinoma, clear cell carcinoma, etc.

ANSWER: Thank you for your critical comments. We added the precise histologic type in the Results section (page 2 and 3, highlighted).

 Minor concerns:

  1. The authors should describe the occasions and first symptoms of the cancers in these patients when the cancers were discovered.

ANSWER: In accordance with reviewer’s kind suggestions, we added the first symptoms of the patients (page 2 and page 3, highlighted).

  1. Has the Case #01 patient been searched for any genetic conditions, including Cowden syndrome/PTEN hamartoma tumor syndrome?

ANSWER: Although we tried to find any symptoms associated with Cowden syndrome or PTEN hamartoma tumor syndrome, we failed to find any symptoms and upper gastroscopies and colonoscopies failed to find any polyp. We also presented the patients’ genetic alterations in Supplementary Table 1.

Reviewer 2 Report

In this presentation of the 2 case series authors nicely describe the role of BAT26 and its association with MSI. They also present two rare cases of BAT26-only instability and conclude they are good candidates for immunotherapy, as they also present high TMB and PD-L1 expression. However, the authors should clarify if TMB and PD-L1 expression are always associated with BAT26 or if this is an incidental finding given the low number of available cases. Thus, do they recommend immunotherapy due to BAT26 or TMB and PD-l1? Moreover, the authors should further explain and describe the potential clinical utility of BAT26 (e.g., could it be used as an alternative to TMB or PD-L1 expression?). 

Author Response

In this presentation of the 2 case series authors nicely describe the role of BAT26 and its association with MSI. They also present two rare cases of BAT26-only instability and conclude they are good candidates for immunotherapy, as they also present high TMB and PD-L1 expression. However, the authors should clarify if TMB and PD-L1 expression are always associated with BAT26 or if this is an incidental finding given the low number of available cases. Thus, do they recommend immunotherapy due to BAT26 or TMB and PD-l1? Moreover, the authors should further explain and describe the potential clinical utility of BAT26 (e.g., could it be used as an alternative to TMB or PD-L1 expression?).

ANSWER: Thank you for your critical comments. Out of 6,476 cases tested for pentaplex MSI-PCR, we identified two BAT26-only instability cancers. Unexpectedly, those two cases were associated with high TMB and high PD-L1 expression. Given that high TMB and PD-L1 overexpression are predictive biomarker for immunotherapy [20], we suggest that although very rare, patients with BAT26-only instability cancer need tests for TMB and PD-L1 and would be a good candidate for immunotherapy. Those descriptions were added in the final manuscript with a new reference (page 5, highlighted).

Round 2

Reviewer 1 Report

The authors responded appropriately to the points raised. However, there is a related issue and a description of it needs to be added. This is a minor concern.

Regarding case #01, endometrioid carcinomas are subdivided into grade 1-3, which differ in their clinical and genetic characteristics. The authors should add a description of the grade or indicate that it is unknown.

Author Response

The authors responded appropriately to the points raised. However, there is a related issue and a description of it needs to be added. This is a minor concern.

Regarding case #01, endometrioid carcinomas are subdivided into grade 1-3, which differ in their clinical and genetic characteristics. The authors should add a description of the grade or indicate that it is unknown.

ANSWER: Thank you for your critical comments. We added the precise grade (grade 2) in the Results section (page 2, highlighted).
